# Living through Cyclone Freddy: "Gendered Health Impacts and Coping Responses in Malawi"

Chipiliro Payesa[1]*, Edith Milanzi[1], Junious Sichali[2], Ruth Holla[1,3], Thelma Kaliu[4]

**1** Femanalytica, Lilongwe, Malawi, **2** Department for Health, University of Bath, Bath, United Kingdom, **3** World Health Organization, Lilongwe, Malawi, **4** GenHub Malawi, Lilongwe, Malawi

* payesac@femanalytica.org

## Abstract

Malawi has increasingly faced severe climate-related disasters, with Cyclone Freddy in 2023 among the most devastating in the country's history. While disasters are often considered gender-neutral events, their impacts are shaped by gendered social roles and structural inequalities, resulting in differentiated experiences among women and men.This study examined the gendered health impacts and the responsiveness of humanitarian interventions during and after Cyclone Freddy in Malawi. We conducted a cross-sectional qualitative study in Blantyre (urban) and Chiradzulu (rural) districts between August and September 2024. Forty-six in-depth interviews were held with survivors, complemented by eight key informant interviews with representatives of governmental and non-governmental organizations involved in the response. Thematic analysis was applied to explore gender-specific experiences, health challenges, and perceptions of response efforts. Findings revealed significant gendered disparities. Women reported heightened loss of reproductive health services, and severe disruption to menstrual hygiene management and exposure to gender-based violence. Maternal health needs were largely unmet, and mental health services were delayed and under-resourced, exacerbating psychological distress among women and girls. Men primarily emphasized loss of livelihoods contributing to psychological distress linked to provider roles. While some humanitarian actors adopted gender-responsive measures, such as dignity kits and separate sanitation facilities, these efforts were inconsistent, often reactive, and limited by funding and planning gaps. Cyclone Freddy underscored the urgent need for gender-sensitive disaster preparedness and response frameworks in Malawi. Integrating gender considerations into health, protection, and recovery strategies is essential to mitigate inequitable impacts across all genders.

**Data availability statement:** Due to the sensitive nature of the qualitative data collected, which includes personal experiences of displacement, gender-based violence, and health challenges following Cyclone Freddy, the full interview transcripts cannot be made publicly available as they contain potentially identifiable and highly sensitive information. This restriction is consistent with the ethical approval granted by the Malawi National Commission for Science and Technology (NCST) and the relevant institutional ethics committees overseeing the study. Data are available upon reasonable request for researchers who meet the criteria for access to confidential data. Requests for access to the data may be directed to the Malawi National Commission for Science and Technology (NCST) by emailing ncrsh@ncst. mw.

**Funding:** This work was supported by the Gender and Public Health Emergencies Small Grant Program (Grant No. 24752 to EM) through a research subgrant awarded to FemAnalytica via Simon Fraser University, funded by the Bill & Melinda Gates Foundation. The funders had no role in study design, data collection and analysis, decision to publish, or preparation of the manuscript. No authors received a salary from the funders.

**Competing interests:** The authors have declared that no competing interests exist.

## Introduction

In March 2023, Malawi experienced one of the most intense and deadly tropical cyclones recorded in the Southern Hemisphere Cyclone Freddy. Cyclone Freddy triggered a state of disaster across 14 districts, displacing over half a million people and resulting in more than 1,000 confirmed deaths, with over 500 people reported missing [1]. Its impact followed closely on the heels of Cyclone Ana in 2022 and Cyclone Idai in 2019, compounding the country's vulnerability to repeated climate-related disasters. These events caused widespread destruction to public infrastructure and disrupted access to essential health services, exacerbating public health risks across affected regions [2].

Climate-related disasters in Malawi have increasingly exposed the deep social and health inequities within emergency response systems. Pre-existing gender inequalities in Malawi shape how individuals experience and recover from crises. Women and girls often face higher risks of gender-based violence, unequal access to economic resources, and limited decision-making power within households and communities. Caregiving responsibilities also disproportionately fall on women, influencing their ability to access services and recovery opportunities. At the same time, men may experience social pressure to fulfil provider roles, shaping how they experience economic loss and psychological distress. These structural and social inequalities form the baseline conditions within which disasters occur, meaning that climate-related shocks such as Cyclone Freddy often exacerbate existing vulnerabilities rather than create them [3].

During Cyclone Freddy, of the more than 2.5 million people affected, 52% were women, including over 100,000 pregnant women, while 65% of the displaced population were female and 55% were children [4]. In previous disasters, such as Cyclone Idai, women also accounted for the majority of those displaced and disproportionately represented household heads among the most affected populations [5]. Research from other global contexts also follow this pattern. Following the 2004 Indian Ocean tsunami, women experienced significantly higher mortality rates than men, not because of biological differences, but due to their disadvantaged social positioning during disaster response [3]. Similarly, limited privacy in shelters, poor sanitation, and weak integration of women's health needs into emergency responses contribute to compounding vulnerabilities—particularly in settings where access to healthcare is already limited [6,7].

The mental health implications of such disasters also remain poorly documented and often neglected in response frameworks. Traumatic experiences, including the destruction of homes, separation from family, and loss of life, heighten risks for depression, anxiety, PTSD, and other psychological conditions. These effects are especially pronounced among women, who are often exposed to additional stressors such as domestic violence and caregiving responsibilities in the wake of such crises [8].

Despite the recognition of these gendered risks, disaster responses continue to be shaped largely by blanket decision-making processes, with limited incorporation of gender-sensitive needs. Effective disaster health responses must therefore go beyond general preparedness to include tailored provisions for different gender groups [9].

The study, using a cross-sectional qualitative approach, aimed to investigate the gender-specific health impacts of Cyclone Freddy as experienced by survivors in Chiradzulu and Blantyre districts in Malawi. In addition, we also examined the gender-responsive health interventions implemented by government and humanitarian organizations in response to the crisis and assessed their effectiveness in addressing the differentiated needs of affected populations with the goal to identify best practices that could inform more equitable and gender-sensitive disaster health response strategies in similar low-resource settings.

Gender in this study is understood as a relational and context-specific construct that shapes the experiences of both women and men within the Malawian context. This study is informed by gender as a social determinant of health framework, which recognizes that gender operates through social norms, roles, power relations, and institutional structures to shape exposure to risk, access to resources, and recovery trajectories. Rather than treating gender as a simple binary variable, this perspective allows for examination of how socially constructed roles around caregiving, provision, and decision-making influence how disasters are experienced and responded to. This framework guided the study design, data collection, and interpretation of findings [10].

## Methods

### Ethics statement

Approval for this study was provided by the National Committee on Research in the Social Sciences and Humanities under the Malawi National Commission for Science and Technology **Protocol no. P.04/24/868**). All participants were adults aged 18 years and above. Written informed consent was obtained from all participants prior to data collection after they were provided with detailed written information about the study. Signed consent forms were securely stored and accessible only to the research team. Participation was voluntary, and participants were informed of their right to withdraw at any time without consequence.

Given the sensitivity of discussions related to gender-based violence, interviewers were trained in ethical interviewing and trauma-informed approaches. Participants who disclosed distress or violence were provided with information on available support services and referral pathways within local health and social support systems.

### Study design and setting

We conducted a cross-sectional qualitative study to examine differentiated health related gender impacts of Cyclone Freddy in Malawi as well as gender responsiveness of the response to the disaster. The study was implemented in two Malawian districts Blantyre and Chiradzulu. These locations were purposively selected to represent both urban (Blantyre) and rural (Chiradzulu) settings, offering a comparative view of the disaster's impacts across different geographical and socio-economic contexts [11]. These districts were also severely affected by Cyclone Freddy.

We conducted in-depth interviews with cyclone survivors from various demographic groups, such as men, women, child-headed households, pregnant women, and adolescents, and key informants from humanitarian organizations that led recovery efforts. These interviews focused on examining the extent to which emergency health interventions addressed gender-specific needs.

### Data collection

Interviews were conducted by trained qualitative researchers affiliated with academic and public health institutions and not involved in humanitarian aid distribution, to reduce potential response bias. Interviews were conducted in private spaces within or near the camps, such as partitioned areas or quiet locations identified by participants, to ensure confidentiality and minimize the risk of conversations being overheard.

A distress protocol was followed during interviews. If participants became emotional, interviews were paused or discontinued based on participant preference, and information about available psychosocial support services was provided.

Data were collected through semi-structured interviews with cyclone survivors and key informants' representatives from governmental and non-governmental organizations.

Access to designated displacement camps was granted through coordination with district authorities and camp management committees. Research procedures aligned with humanitarian research conventions, including principles outlined in the SPHERE standards and ethical guidance for research in crisis settings.

## Data analysis

Audio-recorded interviews were transcribed verbatim and translated from Chichewa to English by experienced researchers with training and experience in qualitative research and public health in Malawi. Analysis was conducted using an inductive thematic analytical approach with NVivo 14 [12] Coding was conducted iteratively by multiple members of the research team. An initial coding framework was developed based on emerging themes and refined through team discussions. Regular meetings were held to review coding consistency and resolve discrepancies. Triangulation across participant groups and key informants enhanced credibility, while detailed documentation of analytic decisions supported transparency. This approach enabled the identification, analysis, and reporting of response patterns, aiding in the organization, description, and detailed interpretation of the information. It was an iterative process in which researchers immersed themselves in the data by reading transcripts multiple times to develop a codebook for data coding.

## Findings

### Participant characteristics

Of the 46 participants, 22 (47.8%) were from Blantyre and 24 (52.2%) were from Chiradzulu districts. The mean age (SD) of the participants was 46.2 (14.6) years in Blantyre and 35.6 (12.9) years in Chiradzulu. The majority were married when the cyclone occurred, and most participants had either attended primary or secondary school education (Table 1).

Table 1. Characteristics of participants by district*.

| Characteristics | Blantyre (N=22) | Chiradzulu (N=24) | Total (N=46) |
|---|---|---|---|
| Age (mean, SD) | 46.2 (14.6) | 35.6 (12.9) | 40.7 (14.7) |
| Sex, female, (n, %) | 12 (54%) | 11 (45%) | 23 (50%) |
| Marital Status, (n, %) | | | |
| Married | 10 (45.5%) | 19 (79.2%) | 29 (63.1%) |
| Widowed | 5 (22.7%) | 2 (8.3%) | 7 (15.2%) |
| Divorced | 3 (13.6%) | 0 (0) | 3 (6.5%) |
| Single | 4 (18.2%) | 3 (12.5%) | 7 (15.2%) |
| Highest Level of Education (n, %) | | | |
| None | 1 (4.6%) | 0 (0) | 1 (2.2%) |
| Primary school | 4 (18.2%) | 17 (70.8%) | 21 (45.6%) |
| Secondary school | 11 (50.0%) | 7 (29.1%) | 18 (39.1%) |
| Tertiary | 6 (27.2%) | 0 (0) | 6 (13.0%) |

*Percentages calculated out of column totals.

## Results

A thematic analysis of the data generated five major themes: Immediate physical health impacts, access to Maternal Health Services including WASH (Water, Sanitation, and Hygiene) and Menstrual Hygiene Challenges, Gender-Based Violence and Long-Term Impacts: Psychological, and Mental Health Consequences.

### 1. Immediate physical health Impacts

The immediate period following Cyclone Freddy revealed pronounced gendered disparities in displacement experiences. Women emphasized the dual burden of physical danger and familial separation. One female participant reflected,

*"While we were trying to escape the water, many people were injured. Some were hit by debris, and others fell while running. There was a lot of confusion and fear." (*Female, 24, Blantyre)

This sense of relational rupture compounded women's emotional distress during evacuation. Beyond separation, women often bore the responsibility of caring for injured relatives even as they struggled to find safe shelter as narrated by one participant.

*"My mother was badly injured in the ribs; she was cut by iron sheets in the ribs, armpits and the body looked like she was beaten. She says she could see the Chanza River close, but she was stuck in the small river and that's what saved her. Also, she was hit by house bricks and rocks as the water was sweeping her away. She was badly injured and to see her now alive, we call it all mercy."*

(Female 27, Chiradzulu)

Men's accounts, while acknowledging similar physical injuries prioritized material loss and livelihood risks. A male respondent described,

"*Houses, I had three houses. All were lost. My house and another semi-detached they were all swept by the water. Everything was lost!... We are just grateful to GOD that we are alive*" (Male, 64, Blantyre).

Such narratives expose how asset destruction exacerbated men's psychosocial trauma, as the loss of homes and tools felt like a loss of personal and economic identity.

### 2. Access to Maternal Health Services

Access to health services in camps following Cyclone Freddy was inconsistent, with gender-specific health needs often overlooked in the early stages. Maternal services were not prioritized, and pregnant women queued alongside others without special consideration. A participant narrated below

*"Regarding my family's well-being, I can say that my wife miscarried after the accident. At that time, it was hard for women to access various services or aid. Like sanitary pads, for instance, it was hard for women to get, and some were on injectable contraceptives. So, during the time of the accident(cyclone), their focus was on healing from all this and forgetting about contraceptives because their heads were saturated."*

(Male 25, Chiradzulu)

Stakeholders acknowledged that their health responses lacked a gender-responsive framework. While some later adjustments, such as the deployment of female health workers, improved access to reproductive care, these were reactive rather than planned. As one key informant concluded,

*"We responded, but not from a gendered understanding of health needs."- Key informant, Humanitarian*

Reproductive health needs were severely disrupted in the aftermath of Cyclone Freddy, with displaced women facing numerous challenges in accessing family planning services, antenatal care, and safe spaces for delivery. These disruptions were compounded by increased risks of gender-based violence (GBV), particularly in overcrowded camps where law enforcement and protection systems were stretched thin.

**WASH (Water, Sanitation, and Hygiene) and menstrual hygiene challenges.** Access to safe and gender-sensitive WASH facilities was severely compromised in the aftermath of Cyclone Freddy. Overcrowded camps, damaged infrastructure, and limited resources created widespread hygiene risks, particularly for women and girls. Shared sanitation facilities lacked privacy and cleanliness, often forcing women to use the same toilets as men, which compromised both dignity and health.

*"There were toilets that were used by both males and females without separating them… some women would go to the toilets while in her period. So, they would stain and thereby compromise hygiene"* (Cyclone Freddy survivor Female, 42, Chiradzulu).

Water scarcity worsened the situation. Unreliable piped water forced reliance on boreholes and open wells, often of poor quality. Basic hygiene tasks became difficult, especially for pregnant women and caregivers. In some areas, toilet facilities were blocked and unusable, contributing to disease risks.

*"The sleeping rooms were okay, but they were not hygienic… some could defecate on the sides, and it became difficult for someone to utilize them"* (Cyclone Freddy survivor Male, 64, Blantyre).

Menstrual hygiene was especially neglected in the early response. Many women lacked sanitary pads, underwear, and private areas to wash and dry cloth substitutes.

*"After the tragedy… I started menstruation while only dressed in a wrapper… I was given sanitary pads, but it was a few days after"* (Cyclone Freddy survivor Female, 25, Chiradzulu).

While some stakeholders acknowledged the lack of gender responsive designation of the toilets and they eventually responded by constructing dedicated women-only toilet blocks, however these improvements came months after the Cyclone Freddy.

*"There were some toilets that were specifically built for women because we understood that… women and girls were the ones who were most affected"* (KII 3). Another key informant expressed how gender needs were accounted for as that is the mandate of their organization *"we do take a gender responsive approach which really aims to address the specific needs and challenges which are faced by the different genders…we provided the menstrual hygiene kits, maternal health supplies which some of the women said they were not enough. (KII 4)"*

While other organizations proactively built separate latrines for women and girls in some camps, female hygiene needs were not necessarily prioritized, and the lack of early planning for menstrual management reflected broader gaps in

recognizing women's unique vulnerabilities. Some camps later distributed hygiene kits, but menstrual products were not always included or sufficient.

### 3. Gender-based violence

Sexual exploitation emerged as a concern across the two camps. Transactional sex, driven by food insecurity and economic desperation, was reported, though these acts were a result of systemic aid failure rather than personal choice. Adolescent girls and young women were particularly vulnerable. Key informants corroborated these accounts, highlighting women engaged in transactional sex as a survival strategy due to a lack of alternative support mechanisms.

> *"Displacement creates conditions where women and girls become more vulnerable to exploitation. When resources are scarce, some individuals take advantage of the situation, which increases risks of gender-based violence in camps."*
>
> *(KII 5)"*

This exploitation exposed major gaps in the humanitarian response, particularly in the protection of displaced women. Predominantly, our findings indicated that organizations, like UNICEF, quickly responded by establishing safe spaces and support mechanisms within the camps. They conducted community education and gender sensitization to empower victims to speak out whenever they encounter violence. However, few incidences of rape and abuse of power were reported in some camps.

> "*Reports of gender-based violence were there; it was mostly happening to girls. Like last time, somebody came to visit his relative, he found a young girl aged fifteen and enticed her by asking her to accompany the man to buy her some food. The girl went with the man, and she was raped. When she came back, she did not disclose it to people at the camp until some days had passed.*"
>
> (Female 45, Blantyre)
>
> *There was rape abuse in the camp. For the chefs especially men at the kitchen, they would entice women to sleep with them in exchange with good food. So, they would tell girls in the camp to have sex with them if they wanted good food like eggs."*
>
> *(Male, 50, Chiladzulu)*

These accounts highlight the urgent need for integrated, gender-sensitive response mechanisms in disaster settings. Key informants emphasized that GBV services must be embedded in emergency response strategies from the outset, not as supplemental considerations, but as essential, lifesaving services.

**Long-term impacts: Psychological, and mental health consequences.** The prolonged aftermath of Cyclone Freddy has deeply affected survivors' mental health, physical well-being, and socio-economic stability. While all survivors endured significant losses, the long-term consequences have manifested in distinctly gendered ways, shaped by social roles, caregiving responsibilities, and access to resources. While most participants acknowledged psychosocial support, most reflected that mental health support was delayed and under-resourced. In the absence of formal psychosocial services, teachers and camp leaders informally identified distressed individuals for referral. However, structured mental health systems were largely absent during critical early periods.

> "*Support came, but it came late, Mental health support was supposed to be offered earlier because people were traumatized … there were no structures for the counsellors,*" (Male, 57, Blantyre).

Mental health challenges extended beyond the individual to entire families. Many women recounted that their children still experience psychological distress during rainy seasons, associating it with the trauma of the cyclone.

"*My last-born still lives in fear during rainy seasons… she got stuck to a wall and survived, but she still lives in fear… the whole family is still psychologically unwell,*" said a mother in Chiradzulu (Female, 38, Chiradzulu).

Another participant described how her firstborn son, who watched the cyclone from a tree, later developed hallucinations:

"*He was hallucinating a lot… we had to send him to stay with relatives so he could forget what happened*" (Female, 29, Chiradzulu).

This may reveal that men's distress frequently stemmed from their inability to fulfil expected roles as economic providers.

Across interviews, participants described enduring psychological distress following Cyclone Freddy, often linked to the traumatic circumstances surrounding the disaster and the disruption of family life. Many accounts highlighted how these impacts persisted long after the immediate crisis had passed. Participants frequently described ongoing fear, anxiety, and behavioural changes among children, particularly during rainy seasons, which were associated with memories of the cyclone. Women emphasized how these psychological effects extended beyond individual experiences to affect entire families, reflecting their caregiving roles and daily proximity to children's emotional well-being. These narratives illustrate how the absence or delay of structured psychosocial services in the early stages of displacement left many families to cope with trauma largely on their own.

## Discussion

This study offers an evidence-based understanding of the gendered health impacts of Cyclone Freddy on affected populations in southern Malawi, informed by both survivor accounts and from frontline humanitarian actors. The findings show gender norms shape not only exposure to risk but pathways to distress and recovery. Women's narratives frequently centred on caregiving responsibilities, disruptions to reproductive and maternal health services, and the emotional toll of maintaining family wellbeing during displacement. In contrast, men's accounts were more closely tied to loss of livelihoods, economic identity, and the psychological strain associated with perceived failure to fulfil provider roles. These findings suggest that disaster impacts are mediated through socially constructed roles, which influence how individuals interpret loss, prioritize needs, and seek support. Understanding these gendered pathways is critical for designing responses that address both health and socio-economic dimensions of recovery. These findings should be understood within a broader gendered system in which socially embedded norms such as expectations around men's roles as providers and women's roles in caregiving shape how individuals experience and respond to crises in the Malawian context. This highlights the importance of contextually grounded and locally informed humanitarian approaches that are responsive to these gendered dynamics.

This study also contributes to growing evidence that disasters amplify pre-existing gender inequalities by providing context-specific insights from Malawi on how gender norms shape both health and socio-economic recovery pathways. By integrating survivor narratives with stakeholder perspectives, the study highlights gaps between gender-responsive policy intentions and implementation in disaster settings.

Study participants reported several immediate effects of Cyclone Freddy, including displacement, injury, deaths, property damage and disrupted access to essential services. These findings are in line with previous studies [13]. Similarly, Braka et al, also reporting on the effects of the 2023 Cyclone Freddy, stated that a total of 659,278 individuals were

displaced in the affected districts, with the majority of them seeking refuge in schools, temporary shelters, health facilities, churches and mosques [9] Elsewhere empirical evidence from natural disasters, such as the 2004 tsunami in India and Indonesia, reported a stark gender disparity in mortality rates, with significantly more women than men dying, [3] These discrepancies are not solely attributable to differences in physical abilities but may be deeply embedded in the gendered dimensions of disaster impact and response

The disruption of health service access following Cyclone Freddy disproportionately affected women, particularly those with chronic illnesses or requiring maternal health services. Women recounted challenges accessing maternal care, as has been reported consistently in other studies, both in Malawi and in other similar settings [14–16]. Even though access to health services was made available in the aftermath of the disaster, women reported avoiding accessing these services for fear of contracting infections in overcrowded places. This was also observed in other studies where the use of maternal and reproductive health services immediately decreased [14]. Men, in contrast, were less likely to cite personal health needs but expressed concern for the health of their spouses and children showing a dual lens where women navigate both personal and caregiving health burdens, while men experience indirect distress through familial health disruptions.

Access to clean water and safe sanitation was a widespread concern, particularly for displaced women. As previously reported by Red Cross on WASH challenges in the aftermath of the 2022 Cyclone Ana and Gombe, which destroyed a total of 337 boreholes, 8 gravity-fed water schemes, 206 water taps and 53,962 latrines, which resulted in the contamination of community wells [13,17]. Whereas in the 2023 tragedy, Cyclone Freddy left 944,784 people with no access to WASH services and destroyed a total of 90,809 latrines [9] After Cyclone Freddy, many latrines in the survivor camps were mixed gender and lacked privacy, causing discouraging use mostly by women. In our study, men's narratives often focused on infrastructure failures and unclean toilets but less on privacy. This contrast also highlights how WASH challenges, while shared, are experienced differently due to gendered social expectations. Similarly, menstrual hygiene management emerged as a significant but often overlooked concern. While some key informants acknowledged this gap and noted efforts to distribute menstrual supplies, these interventions were often delayed.

Both survivor and stakeholder accounts confirmed the occurrence of transactional sex, coercion by aid providers or men within camps after Cyclone Freddy also as reported by UNICEF [18]. Although some men denied security failures, others admitted that policing was inadequate. These perspectives reflect both denial and acknowledgment of GBV's pervasiveness, reinforcing the need for survivor-cantered protection services. The exploitation of women and girls during disasters has been widely reported by other scholars, including Brown S et al, who have reported harassment, trafficking and sexual exploitation after the occurrence of earthquake in Nepal [5].

Another important but less studied aspect of health impacts was mental health. The array of mental health conditions potentially exacerbated by cyclones extends beyond depression to include acute stress disorder, anxiety, adjustment disorders, PTSD, and sleep disorder [8] In our study the psychological trauma following Cyclone Freddy was universal but experienced through gendered lenses. Women frequently spoke of persistent fear, insomnia, and emotional distress, especially when recalling family deaths or the loss of their homes. Men, meanwhile, focused on economic strain, the loss of businesses, and the frustration of not being able to provide. Even though most survivors highlighted requiring mental health services, there was a gap in the provision of mental health services, as most of them did not receive the required [8,19] psychosocial support due to a shortage of trained counsellors. Scarcity of trained counsellors was also observed in previous studies which called for professionally trained counsellors and psychologists to provide mental health services to cyclone survivors as well as survivors of any other disaster [13,17]. The mental health effects of such experiences, combined with structural barriers to recovery highlight the urgency of a gender-informed mental health and livelihoods response.

The patterns observed in participants' narratives also reveal how gender norms influence not only what individuals experience but how they articulate those experiences. Men tended to frame impacts primarily through economic loss and provider roles, with limited discussion of their own health needs, while women more frequently emphasized caregiving responsibilities and access to health services. The relative absence of men's discussions around personal healthcare and

women's limited emphasis on independent livelihoods suggests that social expectations shape both perceived priorities and help-seeking behaviours during crises.

The gender responsiveness of humanitarian actors varied. While some efforts were made, such as providing separate sanitation facilities, recruiting female health workers, key informants admitted that gender was not always integrated into preparedness plans and part of this was attributed to lack of gender disaggregated data. The absence of gender sensitive policies in making disaster-related decisions and ideological barriers can restrict women's access to crucial public shelters for survival, a situation highlighted by the catastrophic cyclone in Bangladesh in 1991 [15]. There was consensus among key stakeholders, however, that future responses must build gender sensitivity into all phases of assessment, planning, implementation, and evaluation.

## Limitations

This study has several limitations. Interviews were conducted several months after Cyclone Freddy, and participants' accounts may therefore be influenced by recall bias. As recruitment occurred primarily within displacement camps, findings may not reflect the experiences of individuals who remained in their communities or sought alternative shelter arrangements. Discussions of sensitive topics such as gender-based violence and reproductive health may also have been shaped by social desirability bias, despite efforts to ensure confidentiality and a supportive interview environment. Additionally, the cross-sectional design captures experiences at a specific point in time and does not account for evolving recovery trajectories. Finally, gender was explored primarily through male and female categories, which may not reflect the full spectrum of gender identities in disaster contexts. Despite these limitations, the study provides important context-specific insights into gendered health impacts in Malawi.

## Policy implications and recommendations

Findings from this study point to specific system gaps that require targeted policy responses. Given participants' reports of disrupted antenatal care, contraceptive access, and maternal triage delays, emergency health deployment should integrate reproductive and maternal health services from the initial phase of camp establishment.

Women's accounts of inadequate menstrual supplies and lack of privacy in sanitation facilities indicate the need for mandatory inclusion of dignity kits and sex-segregated WASH facilities within the first phase of relief distribution.

Men's and women's narratives of livelihood loss and psychological distress underscore the importance of combining gender-inclusive livelihood recovery programs with structured psychosocial services embedded within community-based platforms.

Reports of transactional sex and camp insecurity highlight the necessity of integrating GBV prevention, safe reporting mechanisms, and protection services into disaster response from the outset.

Finally, inconsistencies in coordination described by key informants suggest that district disaster management structures require formalized gender focal leadership and monitoring mechanisms

## Conclusion

The findings of this study demonstrate that Cyclone Freddy's impact in Malawi was neither gender-neutral nor uniformly experienced. Importantly, gaps in humanitarian preparedness and response often amplified these disparities. Beyond short-term aid, long-term strategies must ensure that they are informed by gender specific needs assessment. Future responses must move beyond generalized support to address the differentiated impacts of crises on men and women, with active participation of affected communities in shaping recovery plans.

## Acknowledgments

We thank all the survivors of Cyclone Freddy who participated in this study and shared their experiences. We are also grateful to the local authorities, camp coordinators, and humanitarian actors who facilitated access and supported the fieldwork. Special thanks go to the key informants from government and non-governmental organizations for their valuable insights. We acknowledge the field research team for their dedication and professionalism throughout data collection

## Author contributions

**Conceptualization:** Chipi Payesa, Edith Milanzi.

**Data curation:** Chipi Payesa, Edith Milanzi.

**Formal analysis:** Chipi Payesa, Edith Milanzi, Junious Sichali.

**Funding acquisition:** Edith Milanzi.

**Investigation:** Edith Milanzi.

**Methodology:** Chipi Payesa, Edith Milanzi.

**Project administration:** Chipi Payesa, Edith Milanzi.

**Supervision:** Chipi Payesa, Ruth Holla.

**Writing – original draft:** Chipi Payesa, Edith Milanzi.

**Writing – review & editing:** Chipi Payesa, Edith Milanzi, Junious Sichali, Ruth Holla, Thelma Kaliu.

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
