## [Decision Letter · Decision Letter 0]

18 Jan 2026

PGPH-D-25-03980

Living Through Cyclone Freddy: “Gendered Health Impacts and Coping Responses in Malawi”

Dear Dr. Chipi Payesa,

Thank you for submitting your manuscript to PLOS Global Public Health. After careful consideration, we feel that it has merit but does not fully meet PLOS Global Public Health’s publication criteria as it currently stands. Therefore, we invite you to submit a revised version of the manuscript that addresses the points raised during the review process.

We look forward to receiving your revised manuscript.

Kind regards,

Muhammad Asaduzzaman, MD MPH MPhil

Academic Editor

Journal Requirements:

1. Please provide additional details regarding participant consent. In the ethics statement in the Methods and online submission information, please ensure that you have specified (1) whether consent was informed and (2) what type you obtained (for instance, written or verbal, and if verbal, how it was documented and witnessed). If your study included minors, state whether you obtained consent from parents or guardians. If the need for consent was waived by the ethics committee, please include this information.

i. Please clarify all sources of financial support for your study. List the grants, grant numbers, and organizations that funded your study, including funding received from your institution. Please note that suppliers of material support, including research materials, should be recognized in the Acknowledgements section rather than in the Financial Disclosure.

ii. State the initials, alongside each funding source, of each author to receive each grant. For example: "This work was supported by the National Institutes of Health (####### to AM; ###### to CJ) and the National Science Foundation (###### to AM)."

iii. State what role the funders took in the study. If the funders had no role in your study, please state: “The funders had no role in study design, data collection and analysis, decision to publish, or preparation of the manuscript.”

iv. If any authors received a salary from any of your funders, please state which authors and which funders.

3. Please ensure that your Ethics Statement is available in its entirety at the beginning of your Methods section, under a subheading 'Ethics Statement'.

4. In the online submission form, you indicated that “Data is available upon request”.

3. Uploaded as supplementary information.

Additional Editor Comments (if provided):

Reviewers' comments:

Reviewer's Responses to Questions

Comments to the Author

1. Does this manuscript meet PLOS Global Public Health’s publication criteria? Is the manuscript technically sound, and do the data support the conclusions? The manuscript must describe methodologically and ethically rigorous research with conclusions that are appropriately drawn based on the data presented.

Reviewer #1: Partly

Reviewer #2: Partly

Reviewer #3: Yes

Reviewer #4: Yes

Reviewer #5: Partly

2. Has the statistical analysis been performed appropriately and rigorously?

Reviewer #1: N/A

Reviewer #2: N/A

Reviewer #3: N/A

Reviewer #4: N/A

Reviewer #5: N/A

3. Have the authors made all data underlying the findings in their manuscript fully available (please refer to the Data Availability Statement at the start of the manuscript PDF file)?

Reviewer #1: Yes

Reviewer #2: No

Reviewer #3: Yes

Reviewer #4: Yes

Reviewer #5: No

4. Is the manuscript presented in an intelligible fashion and written in standard English?

Reviewer #1: Yes

Reviewer #2: Yes

Reviewer #3: Yes

Reviewer #4: Yes

Reviewer #5: No

5. Review Comments to the Author

Reviewer #1: This is a very interesting study, valuable new information and has the potential to meet the standards. There are just a few minor but important issues to address. It would be good from the start to push the current knowledge about this issue further and be careful not to conflate 'gender' with 'women and girls.' In some places in the manuscript you highlight gendered different experiences eg. men's emphasis on livelihoods impacts and women's disproporation burden and experience during crises as caregivers. But in other cases, there is a conflation of 'gender' as 'women and girls' which does a disservice to the idea of a rigorous gender analysis (eg. line 21). It would be good for example to identify like issues like men with HIV and women living with HIV. I might guess that there are different gendered experiences in poor treatment access following crisis -- eg. maybe women are normally more likely to get care in prenatal care contexts and given that comprehensive SRH services are often not well prioritized in emergencies then HIV care lacks for them more than men? It would be good to push the analysis further to really drill into these next level differences and not to stop at the 'women and girls'. Another way to do this would be to be more explicit upfront about the baseline inequality or baseline experiences of GBV etc then to more explicitly demonstrate the differential effect after the crisis. Much of the qualitative data does this but adding a little bit more context data or any additional data that would highlight the way the the absence of a gendered response exacerbates the situation. How are men talking about care eg? And how are women talking about livelihoods if at all? If not at all it might be helpful to share that as well.

Overall this is a really important topic, very resonant. It would be even better to be sure you are pushing it beyond what is known in the field (this should also be stated -- eg citing the current state of knowledge -- ie that there is some understanding that crisis experiences and impacts are gendered but perhaps the policies are application of policies are lagging). It does not seem accurate to say up front that it is considered gender neutral. Eg. here as noted 'disasters are gender neutral (meaning everyone gets hit by disasters) "but their impacts are not" https://www.worldbank.org/en/topic/disasterriskmanagement/publication/gender-dynamics-of-disaster-risk-and-resilience#:~:text=Resource%20and%20structural%20constraints%20are,gender%2Ddifferentiated%20impacts%20of%20disasters.

Overall, very important research but could go a bit further in the analysis and framing in order to actually push this area of knowledge forward.

There are numerous spacing issues -- check that the period is right after the sentence end and that there are no extra spaces.

Reviewer #2: Dear Authors,

Thank you for the opportunity to review your manuscript, “Living Through Cyclone Freddy: Gendered Health Impacts and Coping Responses in Malawi.” The study is timely, policy-relevant, and well aligned with global priorities on climate change, gender, and health. The use of survivor narratives and key informant interviews provides rich qualitative insight and addresses an important gap by centering gendered lived experiences in a low-resource setting. However, substantial revisions are required to strengthen its conceptual clarity, methodological transparency, and analytical rigor.

Major Comments to the Authors.

First, the conceptual framing of gender is insufficiently developed. Gender is largely treated as a descriptive binary category rather than as a social and structural determinant shaping vulnerability, access to services, and recovery. The authors should clearly articulate the analytical framework guiding the study and explain how gender theory informed study design, analysis, and interpretation.

Second, the methods section lacks critical detail. The process of participant recruitment within camps, criteria for inclusion, and procedures for ensuring gender balance are not clearly described. The authors should also explain how data saturation was assessed and provide more information on interviewer characteristics, reflexivity, and management of power dynamics, particularly given the sensitivity of topics such as gender-based violence and reproductive health.

Third, the analytical procedures require clarification. While thematic analysis is cited, the manuscript does not adequately describe code development, team-based coding processes, or strategies used to enhance trustworthiness (e.g., triangulation, peer debriefing). Without this information, it is difficult to assess the robustness of the findings.

Fourth, the results section is overly long and at times overly descriptive. The authors should reduce repetition, use quotations more selectively, and strengthen synthesis across themes. Clearer analytical contrasts between women’s and men’s experiences would improve the interpretive depth of the findings.

Fifth, although the discussion engages with relevant literature, it often reiterates results rather than extending analysis. The authors should more clearly articulate how this study advances existing knowledge on Cyclone Freddy and gendered disaster health impacts in Malawi.

Finally, ethical considerations related to disclosures of gender-based violence require more explicit discussion, including referral pathways and participant protection. The policy recommendations would also benefit from being more specific and clearly linked to the study’s empirical findings. A dedicated limitations section is strongly recommended.

In summary, the study has strong potential, but significant revisions are needed to improve conceptual grounding, methodological transparency, and analytical contribution.

Reviewer #3: Please refer to my comments in the attached document regarding necessary revisions in specific areas. Overall, this study is a comprehensive piece that explores various dimensions of cyclone impacts and concern. While these findings may not be entirely new within the target regions, they offer valuable insights and practical evidence that will certainly assist policymakers and aid agencies in refining their programming and priorities.

Reviewer #4: This manuscript is a valuable contribution to the field of public health and disaster response, particularly in low-resource settings. The introduction provides a strong rationale for the study and connects it to broader literature on gender and disaster response. It is well designed, although the findings could benefit from clearer structure by rephrasing and organizing the headings (main themes) and subheadings. The discussion section is quite dense and breaking it into shorter paragraphs would improve flow. Some sentences are somewhat long and could be simplified for clarity. Finally, it could also benefit from mentioning limitations, such as the challenges of conducting research in disaster settings and any potential biases.

Reviewer #5: This qualitative study describes differences in men's and women's experiences during a natural disaster and resulting displacement, as well as experiences with humanitarian assistance in Malawi. The findings contribute the global health literature on the importance of gender-responsive humanitarian assistance and also documenting the unique burdens that populations may experience based on their gender/sex, age, disability status, and so forth.

Overall, the manuscript would benefit from careful copy-editing to address errors in punctuation and formatting.

In the presentation of the Results, some of the thematic statements were out of sequence with the quotes being presented. I recommend that the authors take care to ensure that the organization of summarizing the themes and then presenting the illustrative quotes has a clear and logical flow throughout.

The weakest aspect of the manuscript was the methods section, and I offer several suggestions to improve this section below:

Study design and setting:

- insert "cross-sectional" before "qualitative study" in the opening sentence (Line 91)

- Line 95 in this section, change "comprehensive view" to "comparative view"

Sampling

- in the first sentence, offer a methods reference for purposive sampling procedure

- at Line 105 in this section, add "quota" or "stratified" before the word "sampling" in that sentence

- The use of "saturation" as a tool to determine when to stop recruitment and stop interviewing participants is not always warranted; In many cases, there are logistical constraints such as funding, research assistance, timeline, access to participants, and so forth. In others, there are methodological considerations that guide the decision regarding sampling for saturation, but there are diverse approaches. The authors should discuss - if they are going to describe saturation for sampling - what approach they are using or modeling their approach after, using appropriate references to support their methodological decision-making.

Data Collection

- the authors reference "designated survivor camps" (Line 118); it would be helpful to the readers to understand more about the settings and also the approvals/permissions that were granted to gain entree into the camps and access to the residents as there are conventions regarding research within these displaced populations (e.g., SPHERE standards, etc.)

- there are also guidance for conducting research with populations affected by humanitarian crises that are used in global health, but none are referenced in the methods section. The authors should add this information and indicate whether or not they followed these recommendations (e.g., ERLHA resources).

- The authors are treating "semi-structured interviews" (a type of interview) with "key informant interviews" (a type of participant) as in the same category in Line 119-119. Key informants can be interviewed using semi-structured interviews, so the sentence as written does not make sense. Please revise.

Ethics Statement

- Here is a good place to discuss using recommended procedures for working with populations affected by disasters and displacement

Data Analysis

- More description of the "experienced researchers" is needed here, perhaps in the form of a positionality statement and indicating who among the author team is being included in this group. A statement of positionality or researcher reflexivity should be moved to the Ethics paragraph.

- "thematic analysis" is too generic to describe the process of analysis, and there should be methodological references throughout the Methods sub-sections to indicate that the authors are following accepted methodological procedures form the peer-reviewed literature. Please add more details.

- Add "applied the code book to the data" in Line 147 in that first sentence (The research team ...... then created....)

- Qualitative analysis is not conducted to ensure the "validity of the findings"; what makes more sense here (Line 148) would be a more detailed description of the inter-rater reliability process (whether that was done quantiatively or qualitatively) and the procedures in a way that it could be replicated based on what is provided in the Methods section.

Line 161 - Revise this sentence, which is confusing. The authors created the themes, themes do not "emerge" on their own.

Results

Line 169 - instead of "disparities", what is being described here is actually "differences" for men and women

Line 170 - more context is needed to understand the circumstances of separation during displacement

Line 183 - change prioritized to "spoke about" in this sentence

Lines 196-7 - This sentence would be better placed in the Discussion

Again.... there are many places where the summative statements in the Results and the quotes don't flow well or seem out of order; revise.

Discussion

- Add a paragraph to describe the gender-responsiveness standards for humanitarian assistance, and why this is important to open this section.

6. PLOS authors have the option to publish the peer review history of their article (what does this mean?). If published, this will include your full peer review and any attached files.

Do you want your identity to be public for this peer review? For information about this choice, including consent withdrawal, please see our Privacy Policy.

Reviewer #1: No

Reviewer #2: No

Reviewer #3:  Yes: MD ABUL HASAN

Reviewer #4:  Yes: Abd Arrahman Alomar

Reviewer #5: No

Figure Resubmissions:

---

## [Decision Letter · Decision Letter 1]

14 Apr 2026

PGPH-D-25-03980R1

Living Through Cyclone Freddy: “Gendered Health Impacts and Coping Responses in Malawi”

Dear Dr. Chipi Payesa,

Thank you for submitting your manuscript to PLOS Global Public Health. After careful consideration, we feel that it has merit but does not fully meet PLOS Global Public Health’s publication criteria as it currently stands. Therefore, we invite you to submit a revised version of the manuscript that addresses the points raised during the review process.

We look forward to receiving your revised manuscript.

Kind regards,

Muhammad Asaduzzaman, MD MPH MPhil

Academic Editor

Journal Requirements:

Additional Editor Comments (if provided):

Reviewers' comments:

Reviewer's Responses to Questions

Comments to the Author

1. If the authors have adequately addressed your comments raised in a previous round of review and you feel that this manuscript is now acceptable for publication, you may indicate that here to bypass the “Comments to the Author” section, enter your conflict of interest statement in the “Confidential to Editor” section, and submit your "Accept" recommendation.

Reviewer #1: All comments have been addressed

Reviewer #3: All comments have been addressed

Reviewer #5: All comments have been addressed

2. Does this manuscript meet PLOS Global Public Health’s publication criteria? Is the manuscript technically sound, and do the data support the conclusions? The manuscript must describe methodologically and ethically rigorous research with conclusions that are appropriately drawn based on the data presented.

Reviewer #1: Yes

Reviewer #3: Yes

Reviewer #5: Yes

3. Has the statistical analysis been performed appropriately and rigorously?

Reviewer #1: Yes

Reviewer #3: Yes

Reviewer #5: N/A

4. Have the authors made all data underlying the findings in their manuscript fully available (please refer to the Data Availability Statement at the start of the manuscript PDF file)?

Reviewer #1: Yes

Reviewer #3: Yes

Reviewer #5: Yes

5. Is the manuscript presented in an intelligible fashion and written in standard English?

Reviewer #1: Yes

Reviewer #3: Yes

Reviewer #5: Yes

6. Review Comments to the Author

Reviewer #1: I really appreciate the importance and findings of the study and am neither surprised nor refuting the findings that the consequences of the crisis are gendered. The findings reflect the context. And the context is gendered in the ways that you offer. This part gets lost with a focus on women and girls being more affected (yes gender inequality is real and needs to be considered and addressed in the humanitarian response and other frameworks-- but it also needs to reflect and respond to nuanced local contexts). For that reason it would have been good to see one or two data points about the gender context in Malawi -- without any tropes. Eg. Is there evidence from other studies that men are more focused (and judged) on material vs relational success in society? In a crisis these society norms play out and are valued so the humanitarian response should better understand, respond to and address these nuances. Ways to do this are more investment and more authentic collaboration with Malawian response teams. And to proactively ensure this understanding is reflected in the disaster preparedness.

In short -- very valuable study, findings make sense but just to urge care in terms of gender that gender is not synonymous with women and girls. If accepted that gender norms and roles are all genders and interactions, expectations and consequences are embedded in a local context (which varies across all context).

Adding a sentence or two in the framing and discussion or conclusion to emphasize this point about gendered embedded in the local context would bring this out more.

Thank you for the paper and the opportunity to review.

Reviewer #3: Found most of the key comments addressed

Reviewer #5: The authors have attended to the suggested recommendations and have successfully revised the manuscript. I recommend an acceptance of the revised paper.

7. PLOS authors have the option to publish the peer review history of their article (what does this mean?). If published, this will include your full peer review and any attached files.

Do you want your identity to be public for this peer review? For information about this choice, including consent withdrawal, please see our Privacy Policy.

Reviewer #1: No

Reviewer #3: Yes: Md Abul Hasan

Reviewer #5: No

 Figure Resubmissions:

---

## [Editor Report · Decision Letter 2]

22 Apr 2026

Living Through Cyclone Freddy: “Gendered Health Impacts and Coping Responses in Malawi”

PGPH-D-25-03980R2

Dear Chipi Payesa,

We are pleased to inform you that your manuscript 'Living Through Cyclone Freddy: “Gendered Health Impacts and Coping Responses in Malawi”' has been provisionally accepted for publication in PLOS Global Public Health.

Best regards,

Muhammad Asaduzzaman, MD MPH MPhil

Academic Editor